# Vertical Transmission and Antifungal Susceptibility Profile of Yeast Isolates from the Oral Cavity, Gut, and Breastmilk of Mother–Child Pairs in Early Life

**DOI:** 10.3390/ijms24021449

**Published:** 2023-01-11

**Authors:** Maria João Azevedo, Ricardo Araujo, Joana Campos, Carla Campos, Ana Filipa Ferreira, Inês Falcão-Pires, Carla Ramalho, Egija Zaura, Eugénia Pinto, Benedita Sampaio-Maia

**Affiliations:** 1INEB—Instituto Nacional de Engenharia Biomédica, 4150-177 Porto, Portugal; 2i3S—Instituto de Investigação e Inovação em Saúde, Universidade do Porto, 4200-135 Porto, Portugal; 3Department of Preventive Dentistry, Academic Center for Dentistry Amsterdam, University of Amsterdam and Vrije Universiteit Amsterdam, 1081 LA Amsterdam, The Netherlands; 4Serviço de Patologia Clínica, Departamento de Patologia e Medicina Laboratorial, Instituto Português de Oncologia do Porto Francisco Gentil, 4200-072 Porto, Portugal; 5Escola Superior de Saúde, Instituto Politécnico do Porto, 4200-072 Porto, Portugal; 6Faculdade de Medicina, Universidade do Porto, 4200-319 Porto, Portugal; 7Department of Gynecology and Obstetrics, Centro Hospitalar Universitário de São João, 4200-319 Porto, Portugal; 8Faculdade de Farmácia, Universidade do Porto, 4050-313 Porto, Portugal; 9Centro Interdisciplinar de Investigação Marinha e Ambiental (CIIMAR), Universidade do Porto, 4450-208 Matosinhos, Portugal; 10Faculdade de Medicina Dentária, Universidade do Porto, 4200-393 Porto, Portugal

**Keywords:** opportunistic/pathogenic fungi, mother–child fungal transmission, antifungal susceptibility

## Abstract

Yeast acquisition begins at birth; however, the contribution of the mother on yeast transmission to the offspring and associated resistance is yet to be clarified. The aim of this study was to explore the vertical transmission of yeasts and their antifungal susceptibility profile in early life. Oral, fecal, and breastmilk samples were collected from 73 mother–child pairs four to twelve weeks after delivery and cultured on Sabouraud dextrose agar with chloramphenicol. The isolates were identified by MALDI-TOF MS. The vertical transmission was studied by microsatellite genotyping. Antifungal susceptibility was determined for fluconazole, voriconazole, miconazole, anidulafungin, and nystatin by broth microdilution assay, following CLSI–M60 guidelines. A total of 129 isolates were identified from 53% mother–child pairs. We verified the vertical transmission of *Candida albicans* (n = three mother–child pairs) and *Candida parapsilosis* (n = one mother–child pair) strains, including an antifungal resistant strain transmitted from breastmilk to the gut of a child. Most isolates were susceptible to the tested antifungals, with the exception of four *C. albicans* isolates and one *R. mucilaginosa* isolate. The vertical transmission of yeasts happens in early life. This is the first work that demonstrated the role of the mother as a source of transmission of antifungal-resistant yeasts to the child.

## 1. Introduction

Although a minority, fungi play a key role in regulating a healthy balance between microbes and the host by being involved in a wide panoply of microenvironment-dependent interactions [1,2]. These microorganisms can modify the host physiology and metabolism and are preponderant to train the immune system and its responses against harmful pathogens [3,4]. Therefore, Krum et al. named fungi as potential “keystone species” [5].

Simultaneously, fungi are also considered opportunistic microorganisms and pregnancy and puerperium seem to be particularly vulnerable periods for fungal infections, both for mother and child [6]. During pregnancy, vaginal *Candida* spp. colonization affects 30–40% of women, which may be due to the increased estrogen levels that promote yeast adhesion and penetration in the vaginal mucosa [7]. A study by Sarifakioglu et al. also described oral candidiasis, a *Candida* spp. infection of the oral mucosa, as the second most frequent oral manifestation in pregnant women, when compared to non-pregnant women [8]. After delivery, this tendency seems to remain and Khadija et al. observed that postpartum females are more susceptible to oral *Candida* spp. colonization, which exhibits enhanced virulence characteristics [8,9]. As for the child, it is speculated that colonization by *Candida* spp. occurs in the first hours after delivery [10]. Moreover, the possibility of fungal isolates being transferred from mother to child is supported by few studies [10,11,12]. Concomitantly, oral candidiasis, also known in infants as oral thrush, is a prevalent condition in children especially until the sixth month of life [13,14,15], and secondary fungal infections often occur in children wearing diapers [16].

Moreover, the vulnerability to fungal infections in the puerperium and the increased use of antifungals particularly in high-income countries [17,18] raise questions about whether this leads to antifungal resistance in the mother and whether this is also transferred to the child. Previous studies in bacteria highlighted the role of the mother as a possible source of antibiotic-resistant microorganisms or antibiotic-resistance genes [19,20]. For instance, Zhang et al. detected antibiotic-resistant bacteria in the breastmilk and gut microbiota of infants, prior to antibiotic exposure and major dietary changes, suggesting such source of antibiotic-resistance strains in infants [21]. The existence of a similar pathway for the intergenerational transmission of antifungal resistance is yet to be determined.

These findings underline the importance of understanding the sources of fungal transmission in early life, particularly due to the scarce literature regarding maternal fungal transmission in healthy infants. Therefore, the aim of this study was not only determining the possibility of vertical transmission in early life by identifying and genotyping yeast isolates from distinct body niches in mother–child pairs, but also to assess the antifungal susceptibility profile of the isolates.

## 2. Results

### 2.1. Clinical and Demographic Characterization

The population studied consisted of 73 mothers and their children, including a pair of twins. The demographic information of the mother, including age and educational level, antibiotic use, and oral health habits as well as the child’s antibiotic, antifungal, or probiotic use, history of candidiasis (oral or perianal), type of delivery, child’s suctional habits, or habits potentiating microbial transmission from mother to her child are displayed in Table 1.

### 2.2. Fungal Carriage and Transmission Profile in Mother–Child Pairs

A total of 129 isolates were identified in 53.4% (n = 39) of mother–child pairs. The isolated species per sample type are displayed in Table 2. The overall carriage of the isolated fungi in these samples was 21.6% in saliva of mothers, 24.7% in oral swabs of children, 23.4% in feces of mothers, 26.5% in feces of children, and 6.3% in breastmilk. *Candida albicans* was the most prevalent species in the oral cavity and gut of mothers (68.4% and 53.8%, respectively), whereas *Candida parapsilosis* was the most prevalent in the oral cavity and gut of children and breastmilk (57.1%, 72.7%, and 60%, respectively). The presence of *C. albicans* in the oral cavity of the mothers and children was concordant with its presence in their gut (Kappa statistics; *p* = 0.006 for mothers and *p* < 0.001 for children). Moreover, there was also a concordance for the presence of oral and fecal carriage of *C. parapsilosis* (Kappa statistics; *p* < 0.001) in children samples, but not in the mothers.

A total of five mother–child pairs (6.8% of all pairs) had simultaneous colonization by *C. albicans* and four (5.5% of all pairs) by *C. parapsilosis*. Overall, there was a significant association between the colonization in mother and child among different sites for *C. albicans* and when the child was a carrier (in the oral cavity or gut) and so for the mother (in the oral cavity, gut, or breastmilk). For *C. parapsilosis*, there was no significant association for the simultaneous presence of this species in mother and child.

Regarding associations between the carriage of certain species and maternal and perinatal factors, the presence of *C. albicans* in child oral swabs was associated with the intake of antifungals in the first month of life (Fisher exact test; *p* = 0.04) and with the history of oral candidiasis on the first month of life (Fisher exact test; *p* = 0.05). *C. parapsilosis* in child oral swabs was associated with postpartum hospitalization (Fisher exact test; *p* = 0.03) and the presence of furry animals at home (Fisher exact test; *p* = 0.03).

In the pairs that shared the carriage of same species, the fungal transmission was evaluated by microsatellite genotyping (n = 53 isolates; Table 3. Interestingly, the similarity of genotypic profiles was observed between 60% (n = 3/5) of mother–child pairs for *C. albicans* and 25% (n = 1/4) for *C. parapsilosis* (Table 3). Overall, among the children that were fungal carriers (either in the oral cavity or gut), the vertical transmission rate was 60% (n = 3/5) for *C. albicans* and 5.6% (n = 1/18) for *C. parapsilosis*.

### 2.3. Antifungal Susceptibility Profile of the Isolates

The antifungal susceptibility was tested in all isolates, with the exception of one *C. albicans* and one *Candida dubliniensis* isolates, which were not viable after thawing. A total of 96.1% (n = 122) of the isolates were susceptible to all antifungals. Only 3.9% (n = 5) of isolates presented resistance, namely four *C. albicans* isolates resistant to voriconazole and fluconazole and one *Rhodotorula mucilaginosa* isolate, resistant to all the azoles and anidulafungin (Table 4). Remarkably, three out of four *C. albicans* isolates (75%) were resistant to both voriconazole and fluconazole and were present in the breastmilk (n = 1) and child feces (n = 2) of the same mother–child pair. These three isolates presented the same genotypic profile, most likely having been vertically transmitted. The other resistant *C. albicans* isolate was detected in the fecal sample of the child. Interestingly, these resistant isolates were detected in infants who had not received any antifungal or antimicrobial drugs.

## 3. Discussion

The aim of this study was to understand the role of the mother as a source of vertical fungal transmission in early life, by identifying, genotyping, and evaluating the antifungal susceptibility of fungal isolates recovered from clinical samples of mother–child pairs. We verified the vertical transmission of some maternal yeasts, namely *C. albicans* and *C. parapsilosis* strains, to the child, including one antifungal-resistant strain transmitted from breastmilk to the gut of the child. For *C. albicans*, the presence of this species in both mother and child was concordant between the assessed niches (oral cavity and gut). The majority of isolates (96.1%, n = 122) were susceptible to the tested antifungals, except four isolates of *C. albicans* and one isolate of *R. mucilaginosa*.

Yeast carriage was detected in 53% of the mother–child pairs, with fungal carriage varying according to the niches, ranging from 21.6% to 23.4% in mother saliva and feces, respectively, 6.3% in breastmilk, and 24.7% and 26.5% in child oral swabs and feces, respectively. Overall, for each kind of gastrointestinal tract samples characterized (from mouth and gut), the carriage rates fit within the range reported by previous studies that used culture-dependent techniques [10,22,23,24,25,26,27,28,29]. For breastmilk, although there is a lack of evidence in the literature, the prevalence observed (6.3%) was discrepant to what was reported previously (around 40%) [30]. The diversity among the isolated species, the majority belonging to *Candida*, was similar to previous studies [6,31]. There was a concordance in the carriage of *C. albicans* in the oral cavity and gut of both mothers and children and of *C. parapsilosis* in the oral cavity and gut of children. In fact, a study by Bougnoux et al. [29] reported that 56 out of 234 subjects (within 25 families) were concomitantly colonized by *C. albicans* in the oral cavity and in the intestinal tract. Previously, Sampaio et al. also reported the same strains of *C. albicans* in samples collected from multiple body sites (e.g., rectum and vagina) of the same patient [30]. This finding suggests that the colonization by yeast species may extend to multiple niches of the same person and, in particular, within the gastrointestinal tract mucosa.

Regarding the carriage of *C. albicans*, the presence of this species in mother and child was concordant among the niches and, whenever the child presented *C. albicans* growth, so did the mother. In fact, after microsatellite genotyping, we verified that 60% of the mother–child pairs sharing this species presented the same strain. The transmission of *C. albicans* from mother to child was reported by a few studies, with some authors suggesting that the initial gastrointestinal inoculum of the fungus begins during delivery, with the skin, vagina, and perianal region being other possible sources of *Candida* spp. [10,32]. A study by Filippidi et al. [6] collected vaginal samples from 374 mothers before delivery and oral and rectal swabs from their children after birth. The authors reported that in the 16 neonates with *C. albicans* growth, all isolates were genotypically similar to their mothers, suggesting a vertical transmission [6]. Furthermore, Kondori et al. [25] reported that *C. albicans* was the most frequently isolated species in the gut of children aged from 3 days until 3 years of life (n = 133). Kadir et al. [23] reported a similar finding in a cohort of 300 Turkish children, with *C. albicans* representing 84.8% of the isolated yeasts. Moreover, the carriage of oral *C. albicans* in children in our study was associated with history of oral candidiasis diagnosis and antifungal intake. Oral candidiasis is a prevalent infection in early life, affecting between 4% and 15% of children, and the opportunistic *C. albicans* is often the responsible microorganism for this infection [33].

Interestingly, *C. parapsilosis* was the most prevalent species in the child samples and it was associated with hospitalization and co-habitation with a furry animal. This is in contrary to the majority of studies which most frequently identify *C. albicans* [34,35], although some authors reported that *C. parapsilosis* appears to be an increasingly dominant pathogen in hospitalized children and neonates, being related with horizontal transmission in intensive care units [36,37,38]. A study by Hannula et al. [32] demonstrated that *C. parapsilosis* was the predominant yeast in the oral cavity of healthy children at ages 12 to 24 months and the source was not the mother. Likewise, the presence of *C. parapsilosis* was not concordant between mother and child in our study. The vertical transmission of this species was only observed in 25% (n = 1) of all pairs that presented simultaneous carriage of this yeast, which explains only 5.6% of the detection of this species in children who harbored it. A study from Waggoner-Fountain et al. [39] that assessed the vertical transmission of *Candida* species from mothers (n = 19) to premature neonates (n = 21) verified that while *C. albicans* was mostly vertically transmitted, the mother did not seem to be a reservoir for *C. parapsilosis.* This yeast seems to be mostly horizontally transmitted from environmental sources and, considering our findings, it may have been transmitted during hospitalization in the first month of life [40].

Regarding resistance, 96.1% of all isolates were susceptible to the selected antifungals, except four isolates of *C. albicans*, resistant to both fluconazole and voriconazole, and one isolate of *R. mucilaginosa*, resistant to all azoles and the echinocandin. From 2009 to 2018, the global consumption of antifungals increased with 6.2% each year in upper- and middle-income countries [41]. The increased intake of antifungals has raised concerns over the possible emergence of resistant strains [42]. Some authors describe antifungal resistance in isolates from mother–child pairs, with antifungal susceptibility profiles similar to our study. Filippidi et al. [6] tested the antifungal susceptibility of yeast isolates from mother–child pairs (vaginal isolates from the mother and oral and rectal isolates from the child). The authors verified that all *C. albicans* isolates were susceptible to amphotericin B, but a few isolates from mothers (1.5% to 6% out of 68 isolates) and from children (6.2% out of 16 isolates) presented resistance to the tested azoles (ketoconazole, fluconazole, itraconazole, and voriconazole). Remarkably, we observed the transmission of a resistant strain from breastmilk to the gut of a child. We hypothesize that this may be a mechanism of intergenerational transmission of antifungal resistance, likewise to what was already reported for bacteria [20,21,43]. Parnanen et al. [43] also verified the similarity between mobile genetic elements (MGEs) in the feces of infants with the MGEs in the maternal breastmilk, suggesting that infants inherit the legacy of previous antibiotic consumption of their mother via the transmission of resistance genes. Kozak et al. [20] demonstrated that highly similar isolates of antibiotic-resistant *Bifidobacterium longum*, *Lactobacillus* spp., and *Enterococcus faecalis* were isolated from five mother–child pairs, indicating the transmission of antibiotic resistance from mothers to infants in early life.

This is one of the few works that evaluated the vertical transmission of yeasts by performing a characterization of the cultivable fungal community of the human gastrointestinal tract and breastmilk, both regarding prevalence, genotype, and antifungal resistance [25,44,45]. Additionally, to our knowledge, this is the first work that reported the vertical transmission of a resistant *C. albicans* strain between mother and child through breastmilk. Research on fungi is still limited compared to bacteria, although the number of invasive fungal infections is rising worldwide [46]. Considering the unmet needs in research of clinical mycology, the World Health Organization recently published a list of fungal priority pathogens, with *C. albicans* listed within the “Critical priority group” and *C. parapsilosis* and *Candida tropicalis* listed in the “High priority group” [47]. All these species were found in this study and four *C. albicans* isolates presented a resistant profile to azoles, which highlights the importance of monitoring yeast carriage and their resistance profile from early life. Additionally, efforts should be made to integrate fungal diagnosis in routine care, as well as to create standard operating procedures for clinicians and laboratories to optimize the diagnosis of fungal infections. Moreover, medical professionals should be encouraged to receive training regarding fungal infections and the rational use of antifungal drugs.

One of the main limitations of this study was the low number of isolates recovered from the samples (n = 129 isolates). The fact that the samples were frozen with BHI and glycerol at −80°C may have impaired the growth of cultivable fungi and consequently led to the underestimation of fungal carriage prevalence. Additionally, as this is a cross-sectional study, the fungal load may have fluctuated over time. A longitudinal sampling should be performed in future to obtain more solid fungal prevalence data.

Recently, our research group has suggested a potential relevant role of the mother in bacterial vertical transmission to the child and this study has confirmed this to fungi [48]. Within the limitations of this study, we hypothesize that delivery may be the initial event for fungal vertical transmission, with the transfer of the fungi present on the skin or in vagina and perianal region to the mouth and, subsequently, gut of the child, which may then be perpetuated by breastfeeding and close contact with the mother. This may explain why isolates found in the gastrointestinal tract and breastmilk of the mother are also present in the oral cavity and gut of the child. Moreover, we believe that this may be a mechanism of intergenerational transmission of antifungal resistance in early life.

## 4. Materials and Methods

### 4.1. Sample Collection

A total of 73 participant mother–infant pairs were recruited at the Obstetrics Service of Centro Hospitalar Universitário de São João, Portugal (CHUSJ) and Unidade Local de Saúde de Matosinhos, Portugal (ULSM). The exclusion criteria for the mothers were as follows: age under 18 years old, pre-existing cardiomyopathy, renal disease, chronic obstructive airway disease, active systemic infection, and genetic syndromes. Written informed consent was obtained from all participants or their parents (for infants). The project was approved by the ethics committee CHUSJ since December 2018 (No.94/2018) and by ULSM (86/CE/JAS), with authorization for the reuse of Hospital Clinical Records for Research.

The participants were clinically characterized and their samples were collected four to twelve weeks after delivery. The obstetric (type of delivery, gestational age at delivery) and perinatal (birth weight, admission to the neonatal intensive care unit) outcomes were recorded, as well as smoking, drinking, nutritional, oral hygiene habits, and medical history of the mother. The feeding mode of the child and habits promoting the transmission of microorganisms were also registered.

The mother–child pairs underwent saliva, stool, and breastmilk collection. Maternal unstimulated saliva samples were collected at least one hour after a meal or mouth/tooth brushing, by asking them to passively drool into a 50 mL sterile tube at room temperature. The oral samples from infants were collected by placing a sterile flocked swab (Copan, Brescia, Italy) on their tongue and rotating it for 5 s. The breastmilk was collected either with an electric pump or manually into a 50 mL sterile collection tube, after discarding the first two drops. Maternal stool was collected into a 60 mL sterile fecal collection container, whereas the infant stool was collected directly from the diaper. After collection, all samples were placed on ice. The saliva and breastmilk samples were aliquoted in 1.5 mL sterile microtubes. Fecal samples were aliquoted in 1.5 mL sterile microtubes using a sterile plastic 1 μL loop. The handles of the oral swabs were cut with scissors (after cleaning its blades with 70% ethanol and passing them through a flame) and the swabs were placed in a 1.5 mL sterile microtube. All aliquoted samples were preserved at −80°C with 500 μL of brain–heart infusion (BHI) broth (Biolab^®^, Budapest, Hungary) with 10% glycerol until further processing.

### 4.2. Fungal Identification

A total of 333 samples were collected from the 73 mother–child pairs (n = 73 unstimulated saliva from the mother; n = 72 oral swab from children; n = 61 feces from mother; n = 63 feces from children; n = 64 breastmilk samples). From each sample, 100 μL was inoculated in Sabouraud dextrose agar (SDA; Biolab^®^, Budapest, Hungary) plates supplemented with 5% chloramphenicol and were incubated at 35 °C for 72 h. From each plate, 1–7 colonies presenting different morphologies were selected. The selected colonies were re-isolated for identification by matrix-assisted laser desorption/ionization mass spectrometry (MALDI-TOF MS; Bruker, Massachusetts, United States of America) and, in some cases, the identification was complemented by analytical profile index (API), and on the basis of microscopic morphological characteristics, such as the production of filaments [49].

### 4.3. Microsatellite Selection, DNA Extraction, and Singleplex Amplification

All isolates belonging to the same species with concomitant presence in the same mother and child pair, namely *C. albicans* and *C. parapsilosis* (n = 53), were selected for genotyping studies. Four microsatellites were selected from previous reports for these specific species (*C. albicans* and *C. parapsilosis sensu stricto*) [50,51,52]. The sequences selected for locus-specific amplification are given in detail in Table 5. The DNA of those selected isolates were extracted using the sodium hydroxide method, previously described [53]. DNA (50–250 ng) was suspended in 50 μL of sterile water and stored at −20 °C. For all microsatellite loci, singleplex PCRs were performed with different isolates in order to evaluate locus amplification specificity and to obtain PCR-amplified alleles for sequence analysis. The PCRs were performed using 1 μL of DNA (1–5 ng/μL), 2.5 μL of multiplex PCR master mix (Thermo Fisher, Waltham, MA, USA) and 0.5 μL of each primer (final concentration of each primer: 2 μmol/mL), in a final volume of 5 μL. After a 95 °C pre-incubation step of 15 min, PCRs were performed for a total of 35 cycles, using the following conditions: denaturation at 94 °C for 40 s, annealing at 52 °C for 60 s, and extension at 72 °C for 1 min, with a final extension step of 10 min at 72 °C. The isolates that did not amplify were repeated with an annealing temperature of 50 °C for 60 s.

### 4.4. Sequencing

PCR products were purified with ExoSAP-IT Express PCR Product Cleanup (Applied Biosystems, Waltham, MA, USA). Sequence reaction was performed by 2 min/96 °C pre-incubation followed by 35 cycles with denaturation at 96 °C for 30 s, annealing at 50 °C for 15 s and extension at 60 °C for 3 min. Sequence products were purified with AutoSeq G-50 Dye Terminator Removal kit (Illustra, GE Healthcare, Chicago, IL, USA). The purified sequence products were dissolved in 10 μL Hi-Di Formamide (Applied Biosystems, Waltham, MA, USA) and run on the 3500 Genetic Analyzer (Applied Biosystems, Waltham, MA, USA).

### 4.5. Antifungal Susceptibility Profile

The susceptibility to several antifungals was determined according to the Clinical and Laboratory Standards Institute (CLSI) M60 2020 guidelines [54]. The selected antifungal compounds were diluted in a range of 0.031–16 μg/mL for voriconazole (Pfizer, New York, NY, USA), 0.125–64 μg/mL for fluconazole (Alfa Aesar, Ward Hill, MA, USA), 0.015–8 μg/mL for miconazole (Sigma-Aldrich, St. Louis, MO, USA), 0.015–8 μg/mL for anidulafungin (Pfizer, New York, NY, USA), and 0.015–8 μg/mL for nystatin (Sigma-Aldrich, St. Louis, MO, USA). The minimal inhibitory concentration (MIC) was interpreted following the CLSI M60 2020 breakpoints [54]. For nystatin, isolates were considered resistant for MIC > 2 μg/mL [55]. Moreover, for miconazole, isolates with MIC > 8 μg/mL were considered resistant [56]. *C. parapsilosis* ATCC 22019 and *Candida krusei* ATCC 6258 strains were used as quality control.

All *Candida* isolates were subcultured in SDA medium (at 36 °C for 24 h). Afterward, a suspension of 0.5 McFarland was prepared for each isolate in a sterile saline solution (0.85% NaCl). These suspensions were diluted twice (1:50 and 1:20) in RPMI-1640 medium (Biochrom AG, Berlin, Germany) buffered with 3-(*N*-morpholino) propanesulfonic acid (MOPS, Sigma-Aldrich, St. Louis, MO, USA) to pH 7.0 ± 0.2. Finally, 100 µL of the diluted yeast suspension was added to each well and incubated with 100 µL of the antifungal compound. Two wells were used as a positive and negative control. Microplates were incubated at 36 °C for 24 h (echinocandins) or 48 h (azoles and polyenes), and the opacity of the solution was visually assessed and compared to the positive control. For the azoles and anidulafungin, MIC endpoints were defined as the minimum concentrations that inhibited 50% of the fungal growth [54]. For nystatin, MIC endpoints were defined as the minimum concentrations that completely inhibited fungal growth [54].

### 4.6. Statistical Analysis

All the results are represented as median (interquartile range) or in percentage (%). The statistical analysis was conducted with Statistical Package for the Social Science (IBM^®^ SPSS^®^ Statistics, SPSS Inc., Chicago, IL, USA, 28 version). Categorical variables were described through absolute or relative frequencies (%) and analyzed using Pearson’s chi-square test or Fisher’s exact test when more than 20% of joint events displayed expected counts less than five or even when any joint event presented expected counts less than one. Continuous variables were described using median and interquartile range. Mother–child pairs were considered as matched groups for statistical analysis. Concordance analysis was done by Kappa Statistics to assess the simultaneous presence of the same isolate in matched groups. For all analyses, statistical significance was assumed when *p* < 0.05.

## 5. Conclusions

In summary, this study demonstrates the transmission of maternal *C. albicans* and *C. parapsilosis* strains to the child. Additionally, for the first time, we described the vertical transmission of an antifungal-resistant strain of *C. albicans*. The majority of isolates were susceptible to the tested antifungals, except four *C. albicans* isolates and one *R. mucilaginosa* isolate. Within the limitations of this study, we showed that the mother may have a relevant role in the transmission of yeasts, including of resistant strains, to the child in early life, which may then be perpetuated by breastfeeding and close contact. As research in the field of clinical mycology is still scarce, further longitudinal cohort studies should be performed to support these findings. In the future, it would be crucial to explore the genes responsible for encoding the antifungal resistance of the strains, including to understand the mechanisms that determine their virulence.

## Figures and Tables

**Table 1 ijms-24-01449-t001:** Demographic and clinical information of the mother–child pairs. Results are shown in prevalence (% (n)) or median (interquartile range).

Maternal and Infant Factors	
**Age of mothers (years)**	34.0 [27.5; 40.5]
**Age of children (weeks)**	7.0 [3.0; 11.0]
**Body mass index before pregnancy (kg/m^2^)**	23.2 [16.7; 29.7]
**Maternal diseases**	
Depression	8.8% (n = 12)
Burnout syndrome	0.7% (n = 1)
Anxiety disorder	8.0% (n = 11)
Neurological disorders	0.7% (n = 1)
Dyslipidemia	2.2% (n = 3)
Arterial hypertension	5.8% (n = 8)
Allergy	12.4% (n = 17)
Rhinitis	12.4% (n = 17)
Asthma	7.3% (n = 10)
Chronic bronchitis	0.7% (n = 1)
Heart disease	0.7% (n = 1)
Renal disease	2.2% (n = 3)
Cancer	2.2% (n = 3)
Vision disturbance	19.0% (n = 26)
Hearing problems	0.7% (n = 1)
Migraine	13.1% (n = 18)
Rheumatism	2.2% (n = 3)
Chronic infectious diseases	0.7% (n = 1)
**Maternal vaginal candidiasis diagnosis during pregnancy**	4.1% (n = 3)
**Maternal vaginal candidiasis diagnosis postpartum**	1.4% (n = 1)
**Antifungals used by the mother**	
During pregnancy	4.1% (n = 3)
Intrapartum	0% (n = 0)
Postpartum	1.4% (n = 1)
**Antibiotics used by the mother**	
During pregnancy	12.5% (n = 9)
Intrapartum	46.6% (n = 34)
Postpartum	9.6% (n = 7)
**Oral hygiene habits**	
Dental appointments (n/last year)	2 [0; 4]
Frequency of toothbrushing (n/day)	2 [1; 3]
Use of mouthwash/dental floss/interdental brushes	74.0% (n = 54)
**Child therapy**	
Probiotic	40.3% (n = 29)
Antibiotic	4.2% (n = 3)
Antifungal	6.9% (n = 5)
**Child oral candidiasis diagnosis**	6.8% (n = 5)
**Sex of the child**	
Male	47.1% (n = 33)
Female	52.9% (n = 37)
**Gestational age at birth (weeks)**	39.0 [37.0; 41.0]
**Type of delivery**	
Vaginal	63.8% (n = 44)
Caesarean section	36.2% (n = 25)
**Child postpartum hospitalization**	17.6% (n = 13)
**Breastfeeding**	91.8% (n = 67)
**Child suctional habits**	
Fingers/Hand	19.0% (n = 12)
Pacifier	25.4% (n = 16)
Fingers/Hand and pacifier	55.6% (n = 36)
**Mother licks pacifier**	10.5% (n = 6)
**Mother kisses child’s mouth**	11.3% (n = 8)

n—number of participants.

**Table 2 ijms-24-01449-t002:** Prevalence of each species identified per sample type.

**Phylum**	** *Genus* **	**Mother Saliva**	**Child Oral Swab**	**Breastmilk**	**Mother Feces**	**Child Feces**
	*Species*					
**Ascomycota**	** *Candida* **					
	*Candida albicans*	68.4%	28.6% ^a,b,c^	40.0%	53.8%	18.2% ^d,e^
	*Candida parapsilosis*	10.5%	57.1%	60.0%	30.8%	72.7%
	*Candida dubliniensis*	10.5%	0.0%	nf	nf	nf
	*Candida guilliermondii*	10.5%	4.8%	nf	nf	nf
	*Candida tropicalis*	nf	4.8%	nf	nf	4.5%
	** *Lodderomyces* **					
	*Lodderomyces elongisporus*	nf	4.8%	nf	nf	4.5%
	** *Geotrichum* **					
	*Geotrichum silvicola*	nf	nf	nf	7.7%	nf
**Basidiomycota**	** *Rhodotorula* **					
	*Rhodotorula mucilaginosa*	nf	nf	nf	7.7%	nf

Statistical concordance between the child oral carriage and ^a^ mother saliva (*p* = 0.001); ^b^ maternal feces (*p* = 0.001); ^c^ breastmilk (*p* < 0.001), and between the child fecal carriage and ^d^ mother saliva (*p* = 0.004); ^e^ mother feces (*p* < 0.001). Statistical analysis was done by using Kappa statistics (after Bonferroni correction for multiple comparisons). nf, not found.

**Table 3 ijms-24-01449-t003:** Genotypic profiles and origin of isolates collected on mother and child pairs.

Species	Pair	Individual	Type of Sample (n)	Genotypic Profile
*Candida parapsilosis*	Pair 1	Child	Oral swab (n = 1)	1
			Feces (n = 3)	1
		Mother	Saliva (n = 1)	2
	Pair 2	Child	Oral swab (n = 2)	3
			Feces (n = 1)	3
		Mother	Breastmilk (n = 1)	4
	Pair 3	Child	Feces (n = 1)	5
		Mother	Saliva (n = 1)	6
	**Pair 4**	Child	Feces (n = 1)	7
		Mother	Feces (n = 2)	7
*Candida albicans*	**Pair 5**	Child	Oral swab (n = 3)	8
			Feces (n = 6)	8
		Mother	Saliva (n = 1)	8
			Feces (n = 7)	8
			Breastmilk (n = 2)	8
	**Pair 6**	Child	Oral swab (n = 2)	9
		Mother	Saliva (n = 2)	9
			Breastmilk (n = 1)	9
	Pair 7	Child	Oral swab (n = 2)	10
			Feces (n = 1)	10
		Mother	Saliva (n = 2)	11
	**Pair 8**	Child	Oral swab (=3)	12
			Feces (n = 1)	12
		Mother	Feces (n = 2)	12
	Pair 9	Child	Oral swab (n = 2)	13
			Feces (n = 1)	13
		Mother	Saliva (n = 2)	14
			Feces (n = 1)	14

Mother and child pairs in bold share the same genotypic profile. n—number of isolates.

**Table 4 ijms-24-01449-t004:** Minimal inhibitory concentrations (MICs) and resistance (R) rates of species isolated from mother–child pairs.

		Antifungal								
Species	Voriconazole		Fluconazole		Miconazole		Anidulafungin		Nystatin	
	MIC	R	MIC	R	MIC	R	MIC	R	MIC	R
*Candida albicans* (n = 62)	0.015–>8	6.45% (n = 4)	0.12–>64	6.45% (n = 4)	0.015–2	0%	0.015–0.06	0%	1–2	0%
*Candida parapsilosis* (n = 51)	0.015–0.25	0%	0.5–4	0%	0.25–1	0%	0.5–2	0%	1–2	0%
*Candida tropicalis* (n = 4)	0.03	0%	0.5	0%	0.12–0.5	0%	0.03	0%	2	0%
*Lodderomyces elongisporus* (n = 4)	0.015	0%	0.12–0.25	0%	0.03	0%	0.015–0.03	0%	1–2	0%
*Candida guilliermondii* (n = 3)	0.06–0.12	0%	8	0%	0.25–1	0%	0.5–1	0%	1–2	0%
Other *	0.12–4	33.3% (n = 1) ^#^	0.015–>64	33.3% (n = 1) ^#^	0.015–>8	33.3% (n = 1) ^#^	0.015–8	33.3% (n = 1) ^#^	1–2	0%

* *Candida dubliniensis* (n = 1); *Geotrichum silvicola* (n = 1); *Rhodotorula mucilaginosa* (n = 1). ^#^ The resistant phenotype was observed for *R. mucilaginosa*. n—number of isolates. MIC values are presented in range in μg/mL. R—resistance, values are presented in percentage (%) and isolates number (n).

**Table 5 ijms-24-01449-t005:** Primers used for microsatellite sequences amplification.

Species	Marker	Primer	Reference
*Candida albicans*	CA1	F 5′-ATGCCATG AGT GGA ATT GG-3′ R 5′-AGTGGCTTG TGT TGG GTT TT-3′	Sampaio et al. [52]
	CA3	F 5′-TTGGAATCACTTCACCAGGA-3‘ R 5′-TTTCCGTGGCATCAGTATCA-3′	Sampaio et al. [52]
*Candida parapsilosis sensu stricto*	CP4a	F 5′-GTGTACACCAACCAATCATCG-3′ R 5′-TTGGAGTAACAAGCGCAGAAG-3′	Reiss et al. [51]
	CP6	F 5′-AATGGAGCAGCTACCACTACC-3′ R 5′-TTGGGGTTTGACGTATTGTCAC-3′	Reiss et al. [51]

## Data Availability

Not applicable.

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
