# Peer review of "Vertical Transmission and Antifungal Susceptibility Profile of Yeast Isolates from the Oral Cavity, Gut, and Breastmilk of Mother–Child Pairs in Early Life"

_ijms, 2023, doi:10.3390/ijms24021449_

Round 1
Reviewer 1 Report
Thank you for the opportunity.
Azevedo et al. reported the vertical transmission of pathogenic fungi by investigating the oral, fecal, and breastmilk samples of 73 pairs of mothers and their neonates. Microsatellite genotyping technique was used to confirm the vertical transmission. Susceptibility of the fungal isolates against fluconazole, voriconazole, miconazole, anidulafungin, and nystatin was also determined. This is a good report and only requires minor revisions. Please find my comments below:
1. Line 28. “It was verified…” it makes no sense. Consider to remove ‘was’
2. Line 43. “Therefore, fungi are considered potential “keystone species”” Consider adding “according to a review article” or “named by a review article as…”
3. Table 1. “Oral health habits” is ‘health’ the right term? What about hygiene?
4. Table 2. Pay attention on its table footnote.
5. Line 145. What do you mean by “Curiously,”?
6. Authors have interesting new findings on “transmission of antibiotic resistance from mothers to infants” please highlight this in the abstract.
7. Can authors find the correlation between the antibiotic intake and fungal susceptibility profile?
8. What is the implications of these findings? What are authors recommendations for clinical practitioners?
9. Please recommend further research strategies in the conclusions.
Author Response
Reviewer 1
Thank you for the opportunity.
Azevedo et al. reported the vertical transmission of pathogenic fungi by investigating the oral, fecal, and breastmilk samples of 73 pairs of mothers and their neonates. Microsatellite genotyping technique was used to confirm the vertical transmission. Susceptibility of the fungal isolates against fluconazole, voriconazole, miconazole, anidulafungin, and nystatin was also determined. This is a good report and only requires minor revisions.
- We would like to thank the reviewer for carefully reading our manuscript.
Please find my comments below:
- Line 28. “It was verified…” it makes no sense. Consider to remove ‘was’.
- We would like to thank the reviewer for this suggestion. We removed the “was” from line 31: “We verified vertical transmission of…”
- Line 43. “Therefore, fungi are considered potential “keystone species”” Consider adding “according to a review article” or “named by a review article as…”
- We thank the reviewer for their suggestion. We changed it accordingly (line 47): “Therefore, Krum et al. named fungi as potential “keystone species”.
- Table 1. “Oral health habits” is ‘health’ the right term? What about hygiene?
- We would like to thank the reviewer for raising this question. Indeed, the best term to describe that section of the table is “Oral hygiene habits”. We changed the subhead of the table accordingly.
- Table 2. Pay attention on its table footnote.
- We would like to thank the reviewer for their comment. We altered the format of the table to improve the location of the footnote.
- Line 145. What do you mean by “Curiously,”?
- We thank the reviewer for their question. The word “curiously” introduces a sentence where we describe the fact that 3 out of the 4 Candida albicans isolates that presented resistance to some of the tested azoles belonged to samples from the same mother-child pair, which was something novel. However, to better highlight the relevance of this finding, we decided to change this adverb to “remarkably” (line 175).
- Authors have interesting new findings on “transmission of antibiotic resistance from mothers to infants” please highlight this in the abstract.
- We thank the reviewer for their suggestion. We believe that now the last sentence of our abstract highlights the novelty of our findings (lines 36-37): “This is the first work that demonstrated the role of the mother as a source of transmission of antifungal-resistant yeasts to the child”.
- Can authors find the correlation between the antibiotic intake and fungal susceptibility profile?
- We thank the reviewer for their question. We tested the association between antibiotic and antifungal intake and the fungal susceptibility profile, but no significant association was found.
- What is the implications of these findings? What are authors recommendations for clinical practitioners?
- We thank the reviewer for their questions. We added the following sentences to the Discussion (lines 272-283): “Research on fungi is still limited compared to bacteria, although the number of invasive fungal infections is rising worldwide [46]. Considering the unmet needs in research of clinical mycology, the World Health Organization recently published a list of fungal priority pathogens, with C. albicans listed within the “Critical priority group” and C. parapsilosis and Candida tropicalis listed in the “High priority group” [47]. All these species were found in this study and four C. albicans isolates presented a resistant profile to azoles, which highlights the importance of monitoring yeast carriage and their resistance profile from early life. Also, efforts should be made to integrate fungal diagnosis in routine care, as well as to create standard operating procedures for clinicians and laboratories to optimize the diagnosis of fungal infections. Moreover, medical professionals should be encouraged to receive training regarding fungal infections and the rational use of antifungal drugs.”
- Please recommend further research strategies in the conclusions.
- We thank the reviewer for their suggestion. We added the following sentences to the Conclusion (lines 426-430): “As research in the field of clinical mycology is still scarce, further longitudinal cohort studies should be performed to support these findings. In the future, it would be crucial to explore the genes responsible for encoding the antifungal resistance of the strains, including to understand the mechanisms that determine their virulence.”

Reviewer 2 Report
Review of manuscript ID ijms-2138766 entitled " Vertical transmission and antifungal susceptibility profile of yeast isolates from the oral cavity, gut and breastmilk of mother-child pairs in early life ".
Authors: Maria João Azevedo et al.
Overall, the manuscript submitted for review is interesting and publishable in International Journal of Molecular Sciences. The results are unique and well documented, and the entire manuscript is cohesively written. However, I have a few suggestions to the authors that the paper needs to be updated before it is officially accepted for publication.
My suggestions:
1) Please adjust the citations in the texts in accordance with the requirements of the journal, e.g. Ln 51. The note applies to the entire manuscript.
2) In general, the full names of fungal species are written in the text when used for the first time. After that, they are written only as an abbreviation. Please correct it, e.g. ln 101. Note to the whole work.
3) Please adapt the table 2 to the requirements of the journal.
4) Why is pg 11 blank?
Author Response
Reviewer 2
Review of manuscript ID ijms-2138766 entitled " Vertical transmission and antifungal susceptibility profile of yeast isolates from the oral cavity, gut and breastmilk of mother-child pairs in early life".
Authors: Maria João Azevedo et al.
Overall, the manuscript submitted for review is interesting and publishable in International Journal of Molecular Sciences. The results are unique and well documented, and the entire manuscript is cohesively written. However, I have a few suggestions to the authors that the paper needs to be updated before it is officially accepted for publication.
- We would like to thank the reviewer for their thorough reading of our manuscript.
My suggestions:
- Please adjust the citations in the texts in accordance with the requirements of the journal, e.g.Ln 51. The note applies to the entire manuscript.
- We thank the reviewer for their correction. We have altered the entire manuscript accordingly.
2) In general, the full names of fungal species are written in the text when used for the first
time. After that, they are written only as an abbreviation. Please correct it, e.g. ln 101. Note to the whole work.
- We would like to thank the reviewer for their correction. We have changed the manuscript according to this comment.
3) Please adapt the table 2 to the requirements of the journal.
- We thank the reviewer for their correction. The background color of the table was removed.
4) Why is pg 11 blank?
- We thank the reviewer for this question. We believe there may have been a problem with the formatting of the manuscript because there is text on page 11 regarding the methodology of our work.

Reviewer 3 Report
This is an interesting paper showing evidence for:
1) vertical transmission of some fungal strains from mother to child;
2) some level of transmission of antifungal resistance of some of these strains;
3) occasional possibility of further perpetuation of antifungal resistant strain transmission via breastfeeding.
The results are interesting and worthy of publication, in my opinion, especially given the scarcity of such studies on antifungal resistance (most of the scientific world's attention has been given to antibacterial resistance rather than antifungal).
A few minor observations that will need to be addressed during this revision round:
Line 141: isolate (singular form, not "isolates")
Line 210: please cite here those "few works" that do exist at the time of this publication
Line 265: "73 participants" may be misleading, given that there are 147 human beings (including the pair of twins). Perhaps better rephrase to "73 participant mother-infant pairs"
Line 277: the "oral hygiene habits of the mother" were already mentioned on the previous line (line 276). Perhaps you meant to add something different here on line 277?
Line 285: how many "first drops" were discarded? This is relevant for reproducing the results if one so desires.
Line 316: a concentration of "2 microM/mL" does not exist ("M" means mol/L) It should be either "2 microM" or "2 micromol/mL"
Lines 261-262 and also Conclusion line 378: please enhance the discussion/conclusion by mentioning in either the discussion or conclusion what further experiments could be done in the future to confirm your stated hypotheses, even though at this time this evidence cannot be provided
Author Response
Reviewer 3
This is an interesting paper showing evidence for:
1) vertical transmission of some fungal strains from mother to child;
2) some level of transmission of antifungal resistance of some of these strains;
3) occasional possibility of further perpetuation of antifungal resistant strain transmission via breastfeeding.
The results are interesting and worthy of publication, in my opinion, especially given the scarcity of such studies on antifungal resistance (most of the scientific world's attention has been given to antibacterial resistance rather than antifungal).
- We would like to thank the reviewer for carefully reading our manuscript.
A few minor observations that will need to be addressed during this revision round:
Line 141: isolate (singular form, not "isolates")
- We thank the reviewer for their comment. However, as we are describing four isolates, we believe the word should be plural: “namely four C. albicans isolates resistant to voriconazole and fluconazole and one R. mucilaginosa isolate”.
Line 210: please cite here those "few works" that do exist at the time of this publication
- We would like to thank the reviewer for their suggestion. We have added references of some of the few works that exist in this field (Line 270) (DOI: 10.1093/mmy/myz091, 10.1186/s12884-019-2618-7, 10.6026/97320630010667).
Line 265: "73 participants" may be misleading, given that there are 147 human beings (including the pair of twins). Perhaps better rephrase to "73 participant mother-infant pairs"
- We thank the reviewer for their suggestion. We have altered the text accordingly (lines 302-304): “A total of 73 participant mother-infant pairs were recruited at the Obstetrics Service of Centro Hospitalar Universitário de São João, Portugal (CHUSJ) and Unidade Local de Saúde de Matosinhos, Portugal (ULSM). The exclusion criteria for the mothers were as follows: …”.
Line 277: the "oral hygiene habits of the mother" were already mentioned on the previous line (line 276). Perhaps you meant to add something different here on line 277?
- We would like to thank the reviewer for their correction. The “oral hygiene habits of the mother” is in duplicate. We deleted the repetition (Lines: 314-315).
Line 285: how many "first drops" were discarded? This is relevant for reproducing the results if one so desires.
- We thank the reviewer for their suggestion. We asked the participants to discard the first two drops of breastmilk. That information was added to the text (line 322).
Line 316: a concentration of "2 microM/mL" does not exist ("M" means mol/L) It should be either "2 microM" or "2 micromol/mL"
- We would like to thank the reviewer for their correction. We corrected that unit accordingly (Line 355).
Lines 261-262 and also Conclusion line 378: please enhance the discussion/conclusion by mentioning in either the discussion or conclusion what further experiments could be done in the future to confirm your stated hypotheses, even though at this time this evidence cannot be provided
- We thank the reviewer for their suggestion. As previously answered to Reviewer 1 on point 8, we added the following sentences to the Discussion (lines 272-283): “Research on fungi is still limited compared to bacteria, although the number of invasive fungal infections is rising worldwide [46]. Considering the unmet needs in research of clinical mycology, the World Health Organization recently published a list of fungal priority pathogens, with C. albicans listed within the “Critical priority group” and C. parapsilosis and Candida tropicalis listed in the “High priority group” [47]. All these species were found in this study and four C. albicans isolates presented a resistant profile to azoles, which highlights the importance of monitoring yeast carriage and their resistance profile from early life. Also, efforts should be made to integrate fungal diagnosis in routine care, as well as to create standard operating procedures for clinicians and laboratories to optimize the diagnosis of fungal infections. Moreover, medical professionals should be encouraged to receive training regarding fungal infections and the rational use of antifungal drugs.”
